

# Different distant metastasis patterns based on tumor size could be found in extensive-stage small cell lung cancer patients: a large, population-based SEER study

Jia Li[1,*], Feng Liu[1,*], Haining Yu[2], Chenglong Zhao[3], Zhenxiang Li[4] and Haiyong Wang[5]

[1] Department of Oncology, Longhua Hospital, Shanghai University of Traditional Chinese Medicine, Shanghai, China
[2] Human Resources Department, Shandong Cancer Hospital and Institute, Shandong First Medical University and Shandong Academy of Medical Sciences, Jinan, Shandong, China
[3] Department of Pathology Oncology, Shandong Cancer Hospital and Institute, Shandong First Medical University and Shandong Academy of Medical Sciences, Jinan, Shandong, China
[4] Department of Radiation Oncology, Shandong Cancer Hospital and Institute, Shandong First Medical University and Shandong Academy of Medical Sciences, Jinan, Shandong, China
[5] Department of Internal Medicine-Oncology, Shandong Cancer Hospital and Institute, Shandong First Medical University and Shandong Academy of Medical Sciences, Jinan, Shandong, China
[*] These authors contributed equally to this work.

Corresponding author
Haiyong Wang,
wanghaiyong6688@126.com

## ABSTRACT

**Background.** Small-cell lung cancer (SCLC) is a malignant cancer with the ability to metastasize quickly. The relationship between tumor size and the distant metastasis patterns of Extensive-Stage Small Cell Lung Cancer (ES-SCLC) has not been reported.
**Objectives.** The aim of this study was to determine the different distant metastasis patterns as they related to tumor size in ES-SCLC.
**Patients and Methods.** We used Surveillance, Epidemiology, and End Results (SEER) population-based data collected from 2010 through 2013 to identify 11058 ES-SCLC patients with definite evidence of distant metastases. Multivariate logistic regression analysis was used to demonstrate the association between tumor size and distant metastasis patterns including bone, liver, brain, and lung metastases. Age, race, sex, and N stage were also selected in the logistic regression model.
**Results.** Subtle differences in metastasis patterns were found among patients based on different tumor sizes. Patients with tumors 3–7 cm have a higher risk of bone metastasis compared with those that have tumors ≤3 cm (OR 1.165, 95% CI [1.055–1.287], $P = 0.003$) and patients with tumors ≥7 cm have a higher risk of lung metastasis (OR 1.183, 95% CI [1.039–1.347], $P = 0.011$). In addition, patients with tumors ≥7 cm had a lower risk of brain metastasis and liver metastasis than patients with tumors ≤3 cm (OR 0.799, 95% CI [0.709–0.901], $P < 0.001$; OR 0.747, 95% CI [0.672–0.830], $P < 0.001$). Interestingly, there was no correlation between a larger tumor and a higher risk of metastasis. However, the tumor metastasis pattern did have some correlation with age, gender, race and N-status.
**Conclusion.** The pattern of distant metastasis of ES-SCLC is related to the tumor size and the tumor size is indicative of the metastatic site. Larger tumor sizes did not correlate

with a higher risk of distant metastasis, but the size is related to the pattern of distant metastasis. The study of different distant metastasis patterns based on tumor size and other clinical features (e.g., age, race, sex, and N stage) in ES-SCLC is clinically valuable.

## INTRODUCTION

Small-cell lung cancer (SCLC) is an extremely aggressive malignancy with approximately 31,000 cases diagnosed annually in the United States. It comprises 14% of total lung cancer diagnoses (*Tarver, 2014*). SCLC is a malignant tumor with high metastatic ability and many metastatic sites, of which bone, brain, liver, and lungs are the most common (*Nakazawa et al., 2012*; *Cai et al., 2018*). Many factors affect the metastasis of small cell lung cancer such as tumor size, lymph node involvement, histological subtype, functional status, age, and gender (*Tas et al., 1999*; *Riihimaki et al., 2014*). Our current understanding of the relationship between clinically relevant factors and patterns of distant metastasis is limited; few studies have explored the association between tumor size and the sites of distant metastasis (*Wang et al., 2018a*; *Milovanovic, Stjepanovic & Mitrovic, 2017*; *Wang et al., 2018b*). Advancements in the treatment of SCLC have lagged behind those for non-small cell lung cancer (NSCLC) and other cancers, especially in the development of molecular profiling and targeted therapies (*Byers & Rudin, 2015*). The ability to predict the risk of distant metastasis with clinically relevant factors in SCLC has important implications for the treatment of this disease.

In this study, the Surveillance, Epidemiology, and End Results (SEER) database was used to analyze the relationship between the size of the tumor and the sites of distant metastasis.

## METHODS

### Patient selection

This study was a retrospective study with data obtained from the SEER registry of the US National Cancer Institute (*SEER, 2018*). The SEER database catalogues approximately one-quarter of the cancer patients in the United States and is constantly renewing its data. The SEER*Stat software (SEER*Stat 8.3.5) was used to identify and screen for patient data recorded between 2010 and 2013 that would be appropriate for this study (Fig. 1). Inclusion criteria were as follows: (1) the pathological diagnosis was microscopically-confirmed by biopsy or cytology samples and there was only one primary tumor; (2) the patient was clinically diagnosed as ES-SCLC according to 7# AJCC staging and had a confirmation of distant metastasis (bone, brain, liver, lung) at the time of initial diagnosis (*Rami-Porta et al., 2017*); (3) variables were defined to include age, race, gender, tumor size, and AJCC staging N. Patients with an ambiguous diagnosis or an uncertain site of distant metastasis were excluded. Patients lacking information about variables including age, race, gender, tumor size, AJCC staging N, and metastatic patterns were also excluded.

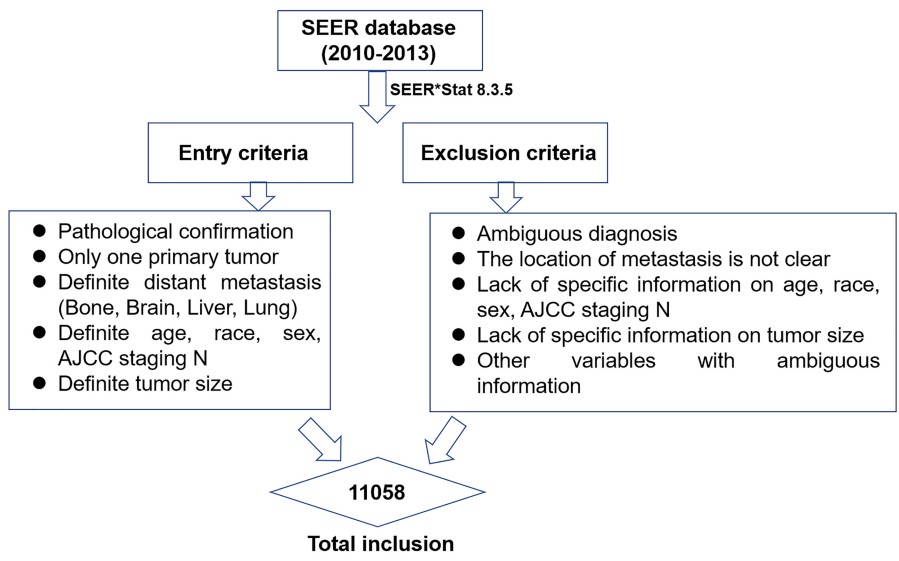

**Figure 1  The flow chart of data selection in this study.**

## Ethics statement

This study was in compliance with the Helsinki Declaration and approved by the Ethics Committee of the Shandong Cancer Hospital. This study was based on the SEER database and did not involve personal privacy information so no informed consent was required.

## Statistical analysis

The variables of age, race, gender, tumor size, N-stage, and metastatic site were included in a multivariate logistic regression analysis to examine the association between clinically relevant factors and specific metastatic patterns. The binomial logistic regression models were used to assess the comparative risk of metastasis. SPSS 22.0 (SPSS, IL, Chicago) was used for data analysis. All statistical tests were bilateral and $P < 0.05$ was considered statistically significant.

## RESULTS

### Patient demographics

From 2010 to 2013, 11,058 ES-SCLC patients were reported in the SEER database; their clinical characteristics are presented in Table 1. 41% of patients were younger than 65 years old, the majority of patients were white (86.4%), and more than half were male. Moreover, the composition of patients with tumor sizes ≤3 cm, 3–7 cm, and ≥7 cm accounted for 23.8%, 47.6%, and 28.6% of patients, respectively. According to the AJCC guidelines for the staging of tumor nodule metastasis (TNM), patients with N2 had the highest proportion of metastasis (57.3%), while patients with N1 had the lowest proportion with only 6.3% of the entire cohort, and N3 patients (24.0%) were in the middle. In addition, the table shows that a total of 3871 (35.0%), 2851 (25.8%), 4956 (44.8%), and 2275 (20.6%) patients

**Table 1** Characteristics of extensive-stage small cell lung cancer from SEER Database from 2010–2013.

| Variables | Number | % |
|---|---|---|
| **Age (years)** | | |
| <65 | 4,536 | 41.0 |
| ≥65 | 6,522 | 59.0 |
| **Race** | | |
| White | 9,555 | 86.4 |
| Black | 1,057 | 9.6 |
| Others | 446 | 4.0 |
| **Sex** | | |
| Female | 5,288 | 47.8 |
| Male | 5,770 | 52.2 |
| **Tumor size(cm)** | | |
| ≤3 | 2,630 | 23.8 |
| 3–7 | 5,269 | 47.6 |
| ≥7 | 3,159 | 28.6 |
| **N stage** | | |
| N0 | 1,375 | 12.4 |
| N1 | 693 | 6.3 |
| N2 | 6,341 | 57.3 |
| N3 | 2,649 | 24.0 |
| **Bone metastasis** | | |
| Yes | 3,871 | 35.0 |
| No | 7,187 | 65.0 |
| **Brain metastasis** | | |
| Yes | 2,851 | 25.8 |
| No | 8,207 | 74.2 |
| **Liver metastasis** | | |
| Yes | 4,956 | 44.8 |
| No | 6,102 | 55.2 |
| **Lung metastasis** | | |
| Yes | 2,275 | 20.6 |
| No | 8,783 | 79.4 |

were diagnosed with bone, brain, liver and lung metastases, respectively. The detailed information is presented in Table 1.

All the possible combinations of metastasis patterns are summarized in Table 2. The results showed that 10.4% of patients had only bone metastases, 12.7% had only brain metastases, 17.6% had only liver metastases, and the proportion of patients with only lung metastases was the lowest, accounting for 6.7% of the total. The most common two-site metastasis was of the bone and liver (11.8%) and the other two-site combination metastases were relatively rare, being those of bone and brain (2.4%), bone and lung (2.0%), brain and liver (3.0%), brain and lung (1.9%), and liver and lung (3.7%). The more common three-site combination metastasis was in the bone, brain, and liver (3.1%) and bone, liver,

**Table 2  Frequencies of combination metastasis sites in ES-SCLC patients.**

| Metastasis | Number | % |
|---|---|---|
| **Only one site** | | |
| Bone | 1,147 | 10.4 |
| Brain | 1,401 | 12.7 |
| Liver | 1,941 | 17.6 |
| Lung | 741 | 6.7 |
| **Two sites** | | |
| Bone+Brain | 263 | 2.4 |
| Bone+Liver | 1,309 | 11.8 |
| Bone+Lung | 224 | 2.0 |
| Brain+Liver | 332 | 3.0 |
| Brain+Lung | 212 | 1.9 |
| Liver+Lung | 406 | 3.7 |
| **Three sites** | | |
| Bone+Brain+Liver | 341 | 3.1 |
| Bone+Brain+Lung | 65 | 0.6 |
| Bone+Liver+Lung | 390 | 3.5 |
| Brain+Liver+Lung | 105 | 0.9 |
| **Four sites** | | |
| Bone+Brain+Liver+Lung | 132 | 1.2 |
| **Others** | | |
| Without (bone,brain,liver,lung) | 2,049 | 18.5 |

and lungs (3.5%). Metastasis to four sites was rare, accounting for 1.2% of the total and without above four sites(bone,brain,liver,lung) accounted for approximately 18.5%.

## Metastasis patterns based on different tumor size

The metastatic sites were identified as bone metastasis, brain metastasis, liver metastasis, and lung metastasis. According to the most recent Eighth Edition of the Tumor, Node, and Metastasis (TNM) Classification of Lung Cancer, tumors ≤1 cm, 1–2 cm, 2–3 cm, 3–4 cm, 4–5 cm, 5–7 cm, and >7 cm are staged as T1a, T1b, T1c, T2a, T2b, T3, and T4, respectively. In the T stage, ≤3 cm is classified as a T1 tumor, 3–7 cm tumors are classified as T2-T3, and tumors ≥7 cm are classified as T4 tumors (*Rami-Porta et al., 2017*). Patients were divided into subgroups according to their tumor sizes of ≤3 cm, 3–7 cm, and ≥7 cm. As shown in Fig. 2A, there are similar proportions of tumor metastasis sites among each tumor size group. The liver was the most common site of metastasis and the lungs were the least common site in all ES-SCLCs. The proportion of liver metastasis was lower in the group with tumors of ≥7 cm (39.8%) than in the other two groups (46.8%). As shown in Fig. 2B, patients with tumor sizes of 3–7 cm were more likely to metastasize regardless of the metastatic pattern. In each distant metastatic site, tumors ≤3 cm and ≥7 cm have a small difference in the proportion of distant metastasis.

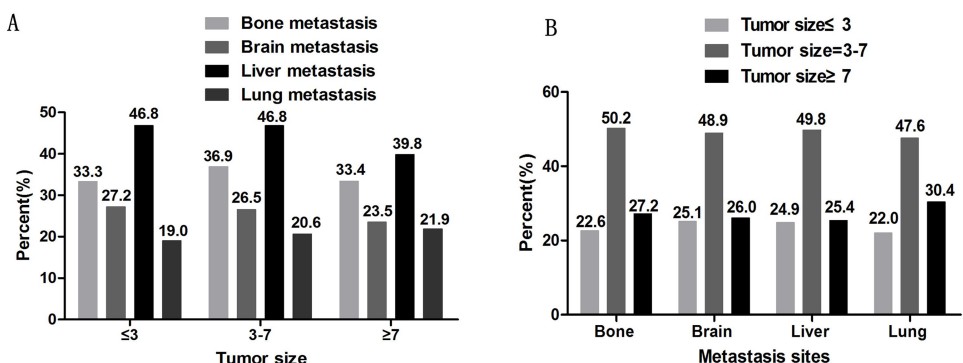

**Figure 2** The percentage of distant metastasis sites. A. The percentage of distant metastasis sites based on different tumor sizes. B. The percentage of tumor sizes based on different distant metastasis sites.

## The association between tumor size and the sites of distant metastasis

We used multivariate logistic regression analysis to analyze the association between metastatic patterns and clinically relevant factors. Age, race, sex, tumor size, and N stage were selected in the logistic regression model and were independent risk factors in the multivariable analysis. Age was not an independent risk factor for liver metastasis ($P = 0.851$), and race and gender were not an independent risk factor for lung metastasis ($P = 0.337$, $P = 0.118$, respectively). Compared with patients with tumors ≤3 cm, there is a higher risk of bone metastasis in patients with tumors 3–7 cm (OR 1.165, 95% CI [1.055–1.287], $P = 0.003$). Patients with tumors ≥7 cm have a higher risk of lung metastasis (OR 1.183, 95% CI [1.039–1.347], $P = 0.011$) but a lower risk of brain and liver metastases than patients with tumors ≤3 cm (OR 0.799, 95% CI [0.709–0.901]) $P < 0.001$; OR 0.747, 95% CI [0.672–0.830], $P < 0.001$). The relationships between the metastasis pattern and age, and gender, race, and N-status were simultaneously observed. Patients ≥65 years old had a lower risk of bone metastasis and brain metastasis ($P = 0.010$, $P < 0.001$, respectively) and a higher risk of lung metastasis ($P < 0.001$). Compared with whites, blacks, and others ethnic patients had a lower risk of bone metastasis and liver metastasis ($P < 0.001$, $P = 0.002$ and $P < 0.001$, $P < 0.001$, respectively) and black races had a higher risk of brain metastasis ($P = 0.004$). We also found that the higher the N stage, the higher the risk of bone and liver metastasis (N1, $P = 0.004$; N2 and N3, $P < 0.001$) and only patients with N3 staging had a higher risk of lung metastasis ($P < 0.001$). However, with a higher N stage, the risk of brain metastasis was reduced ($P < 0.001$) (Table 3).

## DISCUSSION

Small-cell lung cancer (SCLC) is characterized by a rapid doubling time and early, widespread metastasis. The incidence of distant metastasis at the time of the initial diagnosis of SCLC was more than 60% and the most common metastatic sites were the liver, bone, brain, lung, and adrenal glands (*Nakazawa et al., 2012*; *Cai et al., 2018*; *Singh et al., 2013*). The typical treatment of this type of cancer involves small diagnostic biopsies

Li et al. (2019), *PeerJ*, DOI 10.7717/peerj.8163

**Table 3** Multivariate logistic regression analysis was used to evaluate the relationship between tumor size and distant metastasis sites.

| Variables | Bone metastasis | | Brain metastasis | | Liver metastasis | | Lung metastasis | |
|---|---|---|---|---|---|---|---|---|
| | OR (95 %CI) | *P* | OR (95 %CI) | *P* | OR (95 %CI) | *P* | OR (95 %CI) | *P* |
| **Age(years)** | | 0.010 | | <0.001 | | 0.851 | | <0.001 |
| <65 | Reference | | Reference | | Reference | | Reference | |
| ≥65 | 0.899 (0.830–0.975) | 0.010 | 0.703 (0.644–0.767) | <0.001 | 0.993 (0.919–1.072) | 0.851 | 1.209 (1.099–1.330) | <0.001 |
| **Race** | | <0.001 | | 0.005 | | <0.001 | | 0.337 |
| White | Reference | | Reference | | Reference | | Reference | |
| Black | 0.737 (0.641–0.848) | <0.001 | 1.233 (1.071–1.419) | 0.004 | 0.713 (0.625–0.813) | <0.001 | 1.119 (0.959–1.306) | 0.152 |
| Others | 0.711 (0.576–0.878) | 0.002 | 1.196 (0.966–1.481) | 0.100 | 0.690 (0.566–0.841) | <0.001 | 1.055 (0.837–1.329) | 0.651 |
| **Sex** | | <0.001 | | 0.001 | | 0.001 | | 0.118 |
| Female | Reference | | Reference | | Reference | | Reference | |
| Male | 1.282 (1.184–1.388) | <0.001 | 1.152 (1.057–1.256) | <0.001 | 1.152 (1.068–1.243) | 0.001 | 1.077 (0.981–1.182) | 0.118 |
| **Tumor Size(cm)** | | <0.001 | | <0.001 | | <0.001 | | 0.040 |
| ≤3 | Reference | | Reference | | Reference | | Reference | |
| 3–7 | 1.165 (1.055–1.287) | 0.003 | 0.968 (0.871–1.077) | 0.553 | 0.994 (0.904–1.092) | 0.894 | 1.097 (0.975–1.235) | 0.125 |
| ≥7 | 0.965 (0.863–1.079) | 0.531 | 0.799 (0.709–0.901) | <0.001 | 0.747 (0.672–0.830) | <0.001 | 1.183 (1.039–1.347) | 0.011 |
| **N stage** | | <0.001 | | <0.001 | | <0.001 | | <0.001 |
| N0 | Reference | | Reference | | Reference | | Reference | |
| N1 | 1.379 (1.128–1.687) | 0.002 | 1.022 (0.839–1.245) | 0.829 | 1.346 (1.117–1.621) | 0.002 | 1.155 (0.916–1.458) | 0.223 |
| N2 | 1.570 (1.375–1.793) | <0.001 | 0.729 (0.642–0.829) | <0.001 | 1.500 (1.330–1.692) | <0.001 | 1.134 (0.975–1.320) | 0.104 |
| N3 | 2.060 (1.782–2.382) | <0.001 | 0.671 (0.580–0.777) | <0.001 | 1.280 (1.119–1.465) | <0.001 | 1.501 (1.273–1.770) | <0.001 |

and the rare use of surgical resection, which provides insufficient amounts of tumor tissue for translational research, making it difficult to understand the underlying mechanisms of disease progression and metastasis in SCLC (*Qin & Kalemkerian, 2018*; *Oberndorfer & Mullauer, 2018*). An improved knowledge of the risks factors for different metastatic sites would help to properly classify patients with advanced stages of the disease and may serve as a reference for personalized treatment strategies.

The aim of the study was to better understand the impact of different clinically relevant factors on distant metastasis, particularly the association between tumor size and the exact pattern of distant metastasis. Tumor size is one of the major prognostic factors in the staging system for non-small cell lung cancer (NSCLC) (*Giroux et al., 2018*). The prognostic value of the tumor size has been demonstrated in pathological NSCLC and SCLC (*Cai et al., 2018*; *Tas et al., 1999*; *Riihimaki et al., 2014*; *Zhang et al., 2015*; *Zhang, Sun & Chen, 2016*). However, the relationship between the tumor size and distant metastasis pattern of SCLC has not been reported. Bone is the most common site for the distant metastasis of lung cancer. A recent study has shown that patient age (OR = 1.024, $p < 0.001$), the concentration of neuron-specific enolase (OR = 1.212, $p = 0.004$), and histopathological types (OR = 0.995, $p = 0.001$) were the independent risk factors for bone metastasis in patients with lung cancer (*Zhou et al., 2017*; *Niu et al., 2014*). Another study has shown that clinical stage, histology, and the clinicopathological characteristics were related to a higher risk of bone metastasis in patients with completely resected non-small-cell lung cancer (NSCLC) (*Wang et al., 2017*; *Oliveira, Mello & Paschoal, 2016*; *Shabani et al., 2014*). The risk factors for bone metastasis in SCLC were not frequently reported and in addition to age, race, gender, and N stage, the tumor size was also a risk factor for bone metastasis. The higher risk of bone metastasis occurred with the tumors of 3–7 cm. (odds ratio = 1.165, $p = 0.003$) and not with the larger tumor size of ≥7 cm (odds ratio = 0.965, $p = 0.531$). Previous research has examined three steps necessary for lung cancer cells to metastasize to bone: (i) escape from the primary tumor; (ii) entering the circulation; and (iii) colonizing the bone (*Luo et al., 2016*). At the onset of metastasis tumor cells detach from the cell cluster of the primary tumor and are regulated by a series of cell adhesion factors (*Perl et al., 1998*). We hypothesized that tumors with a tumor size >7 cm would not easily detach from the primary tumor to begin the subsequent metastasis. However, the specific molecular mechanisms by which lung cancer cells metastasize to bone still requires further research and exploration. Other research reports that lung cancer is more prone to bone metastases because of the microenvironment of the bone that is affected by the bone matrix, the immune system cells, and the same cancer cells (*Roato, 2014*). At the time of the initial diagnosis, approximately 20% of patients with SCLC have detectable brain metastases (*Eze et al., 2017*), which is roughly in line with our findings. According to the most recent Eighth Edition of the Tumor, Node, and Metastasis (TNM) Classification of Lung Cancer, a larger tumor indicates a higher T stage (*Rami-Porta et al., 2017*). A study showed that high T stage, high neutrophil-to-lymphocyte ratio, early thoracic radiotherapy, and fewer chemotherapy cycles were risk factors for brain metastases (*Zheng et al., 2018*). Our results were inconsistent with those studies and found that patients with tumors ≥7 cm (odds ratio = 0.799, $P < 0.001$) had a significantly lower probability than those with

tumors ≤3 cm to develop brain metastasis. Similarly, a higher N stage correlated with a lower risk of brain metastasis (N2, odds ratio = 0.729, $P < 0.001$; N3, odds ratio = 0.671, $P < 0.001$). SCLC is a tumor that develops brain metastasis very early, typically because small cell lung cancer originates from pulmonary neuroendocrine cells and other potential candidate cells, such as alveolar type 2 cells (*Bunn et al., 2016*). There is also ample opportunity for the seed cells to find a receptive environment (*Lukas et al., 2017*). It is possible that chemokines and adhesion molecules play an important role in lowering the risk of brain metastasis with larger tumor sizes and higher N stages (*Takano et al., 2016*). Other studies have reported that the cumulative years of pack smoking is associated with a greater velocity in brain metastasis (*Shenker et al., 2017*). Due to the limitations of the SEER database specific information on smoking status was not collected. This study did not analyze the relationship between smoking status and brain metastasis.

The liver was the most prevalent site of metastasis (61.9%) and liver metastasis was the most common site of hematogenous metastasis in ES-SCLC (*Ren et al., 2016*). Some studies have demonstrated that liver metastasis may not be associated with the advancement of TNM staging (*Kagohashi et al., 2003*). We analyzed the risk factors for liver metastasis from ES-SCLC patients using race, sex, tumor size, and N stage as independent risk factors in the multivariable analysis. The risk of liver metastasis was lower when the tumor was ≥7 cm, odds ratio = 0.747, $P < 0.001$). Additional research is necessary to identify why patients with tumors ≥7 cm are not prone to develop distance metastasis, especially to the liver. For lung metastasis, we found that age, tumor size, and N stage were independent risk factors and that it was more likely for lung metastasis to occur when the tumor was ≥7 cm (odds ratio = 1.183, $p = 0.011$) and the N stage was higher (N3, odds ratio = 1.501, $P < 0.001$). It is possible that the larger tumor size is associated with a higher probability of lymph node metastasis and local disease extension (e.g., main stem bronchus involvement, visceral pleura invasion, chest wall invasion) (*Chen et al., 2018*). Finally, we observed the relationship between the metastasis pattern and age, gender, race and N-status. Patients ≥65 years old had a lower risk of bone and brain metastasis ($P = 0.010$, $P < 0.001$, respectively) and a higher risk of lung metastasis ($P < 0.001$). Ethnic patients had lower risk of bone metastasis and liver metastasis when compared with whites, blacks, and others ($P < 0.001$, $P = 0.002$ and $P < 0.001$, $P < 0.001$, respectively) and black races had a higher risk of brain metastases ($P = 0.004$). The correlation between these clinical factors and distant metastatic sites may involve differences in population characteristics and further exploration is warranted. We also found that higher N stages (N1, $P = 0.004$; N2 and N3, $P < 0.001$) correlated with a higher risk of bone and liver metastasis. This is consistent with current reports that the volume and number of metastatic lymph nodes are closely related to the site of metastasis (*Zhang, Sun & Chen, 2016*).

There are some limitations to this study. First, it is not possible to assess the impact of chemotherapy, radiation therapy, or smoking status on metastasis because this information is missing from the SEER database. Secondly, the aim of this study was to determine the different distant metastasis patterns based on tumor size in ES-SCLC and we only analyzed the relationship between tumor size and the distant metastatic patterns. When extracting data from the SEER database, we did not extract the corresponding information about

survival rates. Thirdly, this study is a non-randomized study and although our sample size is large, there are inherent defects in any retrospective study. Fourth, the sites of metastasis that were analyzed were limited to the bone, lungs, liver, and brain. Although the common sites of SCLC metastasis are bone, liver, lungs, and brain (*Nakazawa et al., 2012*; *Cai et al., 2018*; *Singh et al., 2013*), metastasis and the combined metastasis of the adrenal gland or other metastatic sites may occur in ES-SCLC patients. Fifth, the tumor size of small cell lung cancer is relatively difficult to measure, so we only selected data with a clear tumor size from the SEER database and any data with an unclear tumor size was not included in this study.

## CONCLUSION

The pattern of distant metastasis of ES-SCLC is related to the tumor size and the tumor size is indicative of the metastatic site. Larger tumor sizes did not correlate with a higher risk of distant metastasis, but the size is related to the pattern of distant metastasis. The study of different distant metastasis patterns based on tumor size and other clinical features (e.g., age, race, sex, and N stage) in ES-SCLC is clinically valuable.

## ACKNOWLEDGEMENTS

We would like to thank the staff of the National Cancer Institute and their colleagues across the United States and those at Information Management Services, Inc., who have been involved with the Surveillance, Epidemiology and End Results (SEER) Program.

### Funding

This study was supported jointly by Special funds for Taishan Scholars Project (Grant no. tsqn201812149), Academic promotion programme of Shandong First Medical University (2019RC004). The funders had no role in study design, data collection and analysis, decision to publish, or preparation of the manuscript.

### Grant Disclosures

The following grant information was disclosed by the authors:
Taishan Scholars Project: tsqn201812149.
Shandong First Medical University: 2019RC004.

### Competing Interests

The authors declare there are no competing interests.

### Author Contributions

- Jia Li performed the experiments, authored or reviewed drafts of the paper, approved the final draft.
- Feng Liu and Haining Yu analyzed the data, prepared figures and/or tables, authored or reviewed drafts of the paper, approved the final draft.

- Haining Yu analyzed the data, prepared figures and/or tables, authored or reviewed drafts of the paper, approved the final draft.
- Chenglong Zhao conceived and designed the experiments, performed the experiments, analyzed the data, prepared figures and/or tables, approved the final draft.
- Zhenxiang Li conceived and designed the experiments, performed the experiments, analyzed the data, contributed reagents/materials/analysis tools, authored or reviewed drafts of the paper, approved the final draft.
- Haiyong Wang conceived and designed the experiments, prepared figures and/or tables, authored or reviewed drafts of the paper, approved the final draft.

### Human Ethics

The following information was supplied relating to ethical approvals (i.e., approving body and any reference numbers):

This study was in line with the Helsinki Declaration and approved by the Ethics Committee of the Shandong Cancer Hospital.

### Data Availability

The raw data is available as a Supplementary File and in the Surveillance, Epidemiology, and End Results (SEER) database (https://seer.cancer.gov/).

### Supplemental Information

Supplemental information for this article can be found online at http://dx.doi.org/10.7717/peerj.8163#supplemental-information.

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
