# Peer review of "Different distant metastasis patterns based on tumor size could be found in extensive-stage small cell lung cancer patients: a large, population-based SEER study"

_PeerJ, doi:10.7717/peerj.8163_

## Round 0.1 · original submission · Major Revisions

Dear authors,

The paper is of interest for the journal but needs to be improved before being re-considered for publication.

Please revise your manuscript according reviewers' suggestions, in particular I recommend you to carefully answer the questions of reviewer #1.

As my personal comment, I would ask you to extend the discussion about implications and limitations of your findings.

Moreover, if available, I would recommend you a language editing by an English native speaker or, at least, a revision of the language through all the manuscript.

Reviewer 1 ·

Basic reporting

The quality of written English is poor and needs significant language editing to facilitate reading.
Abstract:
Abstract needs to be rewritten:
Clearly formulate the aim of the study.
Line 28: you should write “11058 ES-SCLC” instead of “11058 SCLC”.
Line 31: “subtle differences” – please specify compared to what?
Lines 32, 33, 34: The results are scarce and need to be rewritten to better present your findings. Also make sure to specify the units of tumor size.
Line 35: The conclusion does not match your results and it is unclear how you can find difference “between tumor size and distant metastasis patterns”. Logistic regression shows associations, not differences.

Introduction:
Lines 44-45: “fewer study explored about the association between tumor size and the sites of distant metastasis”. Please add references there, and expand on the findings of those previous studies. This is essential to show the gaps in the literature and justify the need for your study.
Line 47-48: Sentence is not clear enough; please rephrase.
Line 50: “...evaluated the effects of several clinical features on the metastasis distribution in ES-SCLC. “ Specify what you meant by “several clinical features“.
The aim was not clearly formulated.
Provide units in all tables and figures.
Raw data is not provided.

Experimental design

In the section on patient selection, you should specify that this was a retrospective study.
Inclusion/exclusion criteria for patients are not well defined.
On which samples was the diagnosis made? (Surgical or biopsy/cytology?)
You need to specify at which time point the presence of metastases was noted (at the time of initial diagnosis or during the entire course of disease).
How many patients had one, two or more metastases? Please specify.
Which type of staging was used? Pathological or clinical TNM classification?
Provide information about the smoking status of patients.
No information about survival was provided. This would be really useful to add to the manuscript.
It is not clear why Ethics Committee of the Shandong Cancer Hospital gives the approval for using US data? Please explain. If there is a number of the ethics approval, specify it in the Methods section.
In the section “Statistical analysis” you wrote that univariate and multivariate logistic regressions were used, but there is no data for univariate analysis in the paper. Moreover, clearly state what was considered as dependent variable and what were potential predictor variables. Specify whether you used binomial or multinomial logistic regression.
In the table 2 (logistic regression) you could have used age as a continuous variable. What was the rationale for binarizing age to two groups (<65 and >65)?
What was the rationale for choosing three tumor size categories (<3, 3-7, >7 cm)?

Validity of the findings

Results are written with a lot of repetitions.
Specify the units of tumor size.
The text in the section “Metastasis pattern on different tumor size” completely repeats the data from Figure 1. This should be avoided.
You wrote that tumors smaller than 3 cm were “unlikely to metastasize”, but your data shows that they metastasized in more than 20%. Please check and correct.
Results of the statistical analysis (logistic regression primarily) are misleadingly presented. For example, the sentence “The results showed that the incidence of patients with tumors 3-7 was significantly higher than patients with tumor 104 size ≤3(OR 1.165, 95% CI 1.055-1.287,P=0.003)” is misleading; namely, logistic regression does not compare the incidences of patients but shows how much higher or lower the risk (odds ratio) of certain event (in this case presence of a particular metastasis) is depending on the specific predictor variable.
There are no comments on the relationship between the metastasis pattern and age, gender, race and N-status.

Discussion:
Line 121: Add citation for “The incidence of distant metastasis at the time of the initial diagnosis of SCLC was more than 60%”. Also add more citations for “most common metastatic sites were the liver, bone, brain, lung and adrenal glands[3]”. I suggest the authors to consult the following recent articles:
https://www.ncbi.nlm.nih.gov/pmc/articles/PMC5541967/
Lines 132-134: This should be moved to the Introduction section.
Line 143-145: Add reference. Moreover, this does not explain why specifically tumors in the size range between 3 and 7 cm spread to bone. Comment on the possible explanation for this observation.
Line 157: Please revise the sentence starting with “The predictive value...“.
Line 167-170: The text “except lung” contradicts the table 2 where it is clear that high N status (N2 or N3) reduce the risk of metastasis only for brain, while N3 increases the risk of lung metastasis (cf. your table 2).
Lines 173-174: Please check the sentence and rephrase for improved clarity.
Line 177-178: Please cite appropriate recent articles, such as:
https://www.ncbi.nlm.nih.gov/pmc/articles/PMC3698258/
https://www.ncbi.nlm.nih.gov/pmc/articles/PMC5541967/

Conclusions:
Other clinical factors (except tumor size) were neglected.

Reviewer 2 ·

Basic reporting

The paper was well writtern. They found that the difference could be found between tumor size and distant metastasis pattern in ESSCLC.

Experimental design

1. Please added the survival analysis in the study.
2. Added the flow chart of data selection.

Validity of the findings

All the results support the conclusion.

Additional comments

1. Make all the limition of SEER database in the disscusion.
2. Clinical tumor size of small cell lung cancer was sometimes difficult to calculate, please add it to the limitation.

---

## Round 0.2 · accepted · Accept

The quality of manuscript is improved, as confirmed by the reviewers. I also appreciated the professional language editing.

Reviewer 1 ·

Basic reporting

The paper has been improved substantially.

Experimental design

The paper has been improved substantially.

Validity of the findings

The paper has been improved substantially.

Additional comments

The authors addressed most comments in a satisfactory way and the paper has been improved.

Reviewer 2 ·

Basic reporting

no comment

Experimental design

no comment

Validity of the findings

no comment

Additional comments

This article was well orgnaized and well writtern.